# Mercury Content and Amelioration of Its Toxicity by Nitric Oxide in Lichens

**DOI:** 10.3390/plants12040727

**Published:** 2023-02-07

**Authors:** Jozef Kováčik, Lenka Husáková, Martina Piroutková, Petr Babula

**Affiliations:** 1Department of Biology, University of Trnava, Priemyselná 4, 918 43 Trnava, Slovak Republic; 2Department of Analytical Chemistry, Faculty of Chemical Technology, University of Pardubice, Studentská 573 HB/D, 532 10 Pardubice, Czech Republic; 3Department of Physiology, Faculty of Medicine, Masaryk University, Kamenice 753/5, 625 00 Brno, Czech Republic

**Keywords:** antioxidants, biomonitoring, heavy metals, reactive oxygen species

## Abstract

Mercury (Hg) content measured in five epiphytic lichen species collected in Slovakia mountain forests ranged from 30 to 100 ng/g DW and was species-specific, decreasing in the order *Hypogymnia* > *Pseudevernia* > *Usnea* > *Xanthoria* > *Evernia prunastri* (but polluted sites had no impact on Hg amount in *Xanthoria*). *Evernia* was therefore used to study the impact of short-term exogenous Hg (100 µM, 24 h) and possible amelioration of Hg toxicity by nitric oxide (NO) donor sodium nitroprusside (SNP). NO was efficiently released from SNP as detected by two staining reagents and fluorescence microscopy and reduced Hg-induced ROS signal and absorption of Hg by thalli of *Evernia prunastri*. At the same time, NO ameliorated Hg-induced depletion of metabolites such as ascorbic acid and non-protein thiols, but not of free amino acids. The amount of metabolites, including soluble phenols, was reduced by excess Hg per se. On the contrary, NO was unable to restore Hg-stimulated depletion of chlorophyll autofluorescence but mitigated the decline of some macronutrients (K and Ca). Data confirm that accumulation of Hg in the epiphytic lichens is species-specific and that NO is a vital molecule in *Evernia prunastri* that provides protection against Hg-induced toxicity with considerable positive impact on metabolic changes.

## 1. Introduction

Mercury (Hg) is one of the most toxic metals due to its specific physicochemical properties, long distance transport in the atmosphere, toxicity even at low concentrations, and considerable accumulation in vascular and non-vascular plants [1,2,3,4,5]. Lichens are not plants, but lichenized fungi with their body (thallus) typically formed by the symbiotic association between fungus and green alga. Despite more common works focused on the use of lichens for biomonitoring of metallic pollution including the occurrence of Hg [6,7,8,9,10,11], physiological studies are rare [12,13]. 

The presence of heavy metals stimulates excessive production of reactive oxygen species (ROS) which is often metal-specific [14,15] and concentration-dependent, also in non-vascular plants and lichens [12,16]. To cope with ROS overproduction, all cells use mainly non-enzymatic antioxidant ascorbic acid and thiol metabolites, and its significance under metal excess is also known in lichens [17]. Phenolic metabolites as additional antioxidants and amino acids as basic metabolic/enzymatic components are also affected by the excess of metals in lichens [18], but the impact of Hg and regulation of metal-induced ROS formation is not well understood.

Nitric oxide (NO) is a unique simple gaseous molecule, but with considerable impact on plant physiology and metabolism, including adaptations to (heavy) metals [19,20]. It was observed—using NO modulators (donors or scavengers)—that NO often improves metal and Hg-induced negative changes in plants [16,21,22,23]. However, the role of NO in lichens is less known and only a few studies have reported the impact of NO scavenger under metal excess [24,25]. Sodium nitroprusside is the most stable and most commonly used NO donor, which may have eventual side effects but is physiologically useful [26], and, due to the low impact of Hg on NO amount observed in the preliminary experiment, NO donor SNP was utilized instead of NO scavenger. Owing to limited data related to SNP/NO effects in lichens, its eventual ameliorative effect on Hg-induced toxicity in the epiphytic lichen *Evernia prunastri* was tested in a laboratory experiment. At the same time, five epiphytic lichens collected throughout Slovakia were analyzed for Hg content to study inter-specific variability and the impact of eventual pollution source on Hg accumulation. 

## 2. Results and Discussion

### 2.1. Hg Content in Natural Epiphytic Lichens 

Analysis of Hg content in natural samples of five epiphytic lichens (originating from forest areas, although close to urban settlements in the High Tatras, Staré Hory, or Ochtinská Aragonite Cave) revealed a lower amount in *Evernia prunastri* than in *Pseudevernia furfuracea* or *Hypogymnia physodes* (two sites for each species), and the same is true when only the High Tatra Mountains locality is compared among all species (Table 1), supporting species-specific difference. Consistent with our findings, *Evernia* accumulated less Hg than *Xanthoria* in a field study from Italy [11]. Also, a laboratory study focused on the uptake of elemental gaseous Hg by lichens showed the content of Hg in the order *Pseudevernia* > *Xanthoria* > *Evernia* [27], which is perfectly in agreement with our data from High Tatra Mountains (Table 1). 

Sites with presumed metal contamination were only available for *Xanthoria* (Párnica and Dolná Streda) and comparison with a potentially cleaner site (High Tatra Mountains) did not show a clear trend, indicating that Hg deposition has no connection with a metallurgical factory or metal dump, at least in the case of *Xanthoria*. A similar result was found in *Xanthoria* from Turkey, where urban samples did not contain more Hg than those from rural areas [28]. It was also observed that Hg amount in all species (in a range ca. 30–100 ng/g DW) was similar or lower than values reported, e.g., in *Usnea* from Antarctica (93–199 ng/g DW or 228–387 ng/g DW [6,9]), *Hypogymnia* from a forest area in Czech republic/Poland (mean value 129 ng/g DW, [29]), *Parmelia* from South Africa (60–218 ng/g DW, [7]), *Xanthoria* from Italy (50–400 ng/g DW, [8]) and *Parmelia* from Italy (142–624 ng/g DW, [10]), indicating that Hg air pollution is probably not problematic in Slovakia.

### 2.2. Impact of Exogenous NO Donor on Hg Uptake and Macronutrients

Owing to the low Hg amount in *Evernia prunastri* mentioned above and its soft thalli suitable for biochemical extraction, we selected this species for subsequent investigation focused on the toxicity of exogenous Hg and cross-talk with nitric oxide (see representative photo of *Evernia* thalli in Appendix A). Total Hg content was not affected by SNP co-application, with an average value of 2.4 mg Hg/g DW, which is similar to data reported by Vannini et al. [13] who also found over 2 mg Hg/g DW in the same species after 1 h of immersion in 100 µM Hg solution, as did we (Figure 1). Lichen *Ramalina farinacea*, with similar fruticose thalli exposed to 100 µM Cd under identical exposure conditions, also contained close to 2 mg Cd/g DW [24], while terrestrial lichens *Cladonia* and *Peltigera*, exposed to 100 µM Hg under similar exposure conditions, contained over 1 mg Hg/g DW [12]; therefore, the sorption of metals seems to be affected by the applied concentration mainly. Theoretical calculations reveal that *Evernia* absorbed only 12% from the available amount (50 mg DW x 50 mL of 100 µM Hg exposure solution = 20.059 mg Hg/g DW versus 2.4 mg Hg/g DW mentioned above). On the contrary, green microalga *Coccomyxa* exposed to 100 µM Hg accumulated 36.5 mg Hg/g DW [14], indicating a much higher sorption efficiency than lichens.

As opposed to total Hg accumulation, higher SNP dose significantly depleted absorbed Hg by 25.8% (Figure 1), indicating that NO released by SNP affects movement of Hg ions within thalli. SNP also suppresses Hg accumulation in vascular plants such as rice where 100 µM Hg combined with the same or higher SNP dose has been shown to lead to lower Hg amount in both shoots and roots [21]. No similar data are available for lichens. 

Hg excess strongly depleted the amount of potassium in thalli, followed by less intensive depletion of Ca and Mg (Appendix A). Similarly, strong 100 µM Hg-induced K depletion (by ca. 95%) has also been observed in the green microalga [14], suggesting that a given dose of Hg is similarly toxic and thus suitable for studying the protective effect of NO in a short-term experimental setup. Co-application of higher (100 µM) SNP dose slightly but significantly improved K status and more visibly Ca status, but had no effect on Mg amount (Appendix A), suggesting that the interaction between Hg, NO, and macronutrients is metal-specific. Comparison of total and absorbed fraction of individual macronutrients showed that ca. half of K and Ca (but not of Mg) is surface-bound and the same, along with the highest absolute amount of Ca, was observed in another epiphytic lichen, *Ramalina* [24].

### 2.3. NO Donor Reduces Hg-Induced ROS Formation 

Induction of ROS formation is metal- and concentration-dependent in lichens, as observed, e.g., in two terrestrial *Cladonia* species exposed to Cu or Cr where also mycobiont/photobiont localization was visualized using given staining reagents [17]. The microscopic photos presented herein show strong ROS induction in response to Hg (see −Hg/0 SNP versus +Hg/0 SNP) using the reagent CM-H_2_DCFDA; a mycobiont is also visible (empty black areas are algal colonies, see Figure 2). The same ROS stimulation has been observed in the green alga exposed to 100 µM Hg, but not Pb [14], or in plant roots exposed to Hg [30], providing evidence that Hg is more toxic than many other common metallic contaminants. It is also known that oxidative stress markers differ in lichens treated by metals in relation to morphology and even depleted H_2_O_2_ content was observed after spray application of Cd [31], while immersion in Cd-enriched solutions led to enhanced ROS or H_2_O_2_ formation detected by fluorescence microscopy and standard spectrophotometry [18]. 

SNP had no impact on the ROS formation and 100 µM SNP (unlike 10 µM SNP) suppressed Hg-induced ROS signal strongly (Figure 2). These qualitative data indicate that even 100 µM SNP had no toxic impact and provides protection against Hg-induced ROS overproduction: this protective effect of SNP was confirmed by elevated signal of the nitric oxide (NO), as detected using two staining reagents with various chemistry (DAN and DAF-FM DA). The NO generation in the lichen *Ramalina* has previously been proven through usage of similar reagents, along with stronger 100 µM Cd-induced ROS signal [24], suggesting similar action of Cd and Hg if identical concentrations are applied. We also note that the NO signal, using DAF FM-DA reagent, probably highlighted the mycobiont (as mentioned above for CM-H_2_DCFDA reagent) and, in agreement, a similar basal reagent (DAF) gave NO signal only in fungal hyphae of the lichen *Ramalina lacera* [32].

In contrast to the present data, where Hg had no stimulatory or rather negative effect on the NO signal (Figure 2, upper panels), the aforementioned *Ramalina* generated more NO in response to Cd excess [24]. The opposite design, i.e., the use of a NO scavenger instead of a donor, revealed that the NO scavenger (with common abbreviation cPTIO) suppresses metal-induced changes of the NO signal and elevates the ROS signal, providing indirect evidence about the protective role of NO [24]. Only limited data are available in lichens, and it has been reported that the effect of cPTIO on the ROS formation in *Ramalina* thalli exposed to rehydration with Pb excess is time-dynamic with limited responses after 4-h post-treatment [25], though we observed clear responses after 24 h. Owing to known reactivity between NO and ROS, a lower NO signal (−Hg/SNP versus +Hg/SNP treatment, Figure 2) may indicate its consumption for the reduced ROS formation. Moreover, Hg-induced suppression of chlorophyll autofluorescence, irrespective of SNP dose (Figure 2), shows that NO may reduce Hg-induced oxidative stress but is unable to reverse all parameters. In agreement with available literature, Hg doses over 50 µM considerably depleted chlorophyll *a* and/or *b* and photosynthetic efficiency in the lichens *Cladonia* and *Peltigera* [12]. In a previous study, Hg also negatively affected proteomic parameters of *E. prunastri* related to photosynthesis [33], which confirms our observations of chlorophyll autofluorescence (Figure 2).

### 2.4. Metabolic Changes: The Protective Role of a NO Donor against Hg Is Not Universal

Ascorbic acid (AsA) and thiol metabolites are essential cellular antioxidants. Their considerably lower accumulation in Hg-exposed thalli (Figure 3) may be a reason for Hg-enhanced ROS formation (Figure 2); the same was observed in algae, also exposed to 100 µM Hg but not to Pb [14]. Higher toxicity of Hg may also be deduced from the fact that 100 µM Cd in the similar fruticose lichen *Ramalina* stimulated accumulation of AsA [24], while the foliose lichen *Hypogymnia* responded to 100 µM Cd excess by depleting AsA and thiols [18]. The negative effect of Hg is concentration- and species-dependent, and no changes in AsA and/or thiols were observed in vascular species *Medicago* treated with 30 µM Hg for 24 h [2] or in *Arabidopsis* leaves treated with 10 µM Hg for 72 h [22]; but even increased accumulation was observed in the cyanobacterium *Nostoc*, exposed to 0.5–3 µM Hg for 72 h [1] or in rice roots exposed to 100 µM Hg, as we also used [3].

SNP alone had no impact on the amount of AsA and thiols, so it does not affect oxidative balance, as was also visible from the fluorescence microscopy (Figure 2). It was a positive finding that SNP ameliorated Hg-induced depletion of these metabolites (Figure 3), so they can contribute to lower oxidative stress or depleted ROS signal (Figure 2). SNP restored 10 µM Hg-induced depletion of AsA and thiols in wild-type *Arabidopsis* roots (which were directly in contact with the treatment solution), but not in leaves [22]. Cross-talk between SNP and heavy metal Cd led to an increase in AsA amount over single treatments in the green microalga [16], and cross-talk between SNP and Pb lowered oxidative stress markers and enhanced the antioxidant system in rice [23], indicating a similar positive effect of SNP in various plant lineages. 

Phenolic metabolites (soluble phenols) remained unaffected by SNP application alone or in combination with Hg, as previously observed in the green microalga [16], but were depleted by more than 50% due to Hg excess (Figure 3). This is a further indication of higher toxicity of Hg alone, as, e.g., 100 µM Cd had no effect on phenols in the foliose lichen *Hypogymnia* [18], and may also indicate specific interaction of Hg with specific lichen metabolites, many of which are (chemically) phenolic in nature. Amino acids, as a basic part of proteins, are often affected by excess metals in plants and lichens and were strongly depleted by the excess Hg (Figure 3), with similar depletion also evoked by the 100 µM Cd excess in *Hypogymnia* [18]. In the green microalga, proteins were strongly depleted by 100 µM Hg excess [14]. SNP had no impact alone, but, contrary to expectations, enhanced Hg-induced depletion of free amino acids (Figure 3). This may indicate that specific interaction between Hg and SNP modulates final response, but no relevant data was found to discuss in detail. We can only speculate that, owing to the complex chemistry of SNP, more SNP was consumed for NO donating in the presence of Hg and this might produce more residue of SNP, leading to this side effect. However, the impact of SNP is not universally positive at the level of all metabolites. To identify metal-specific effect, 100 µM Hg was compared with 100 µM Cu or Cd; data revealed that only Hg depleted phenols, but Hg and Cu depleted free amino acids (Appendix A), suggesting that the effect of Hg is more negative than that of Cd.

### 2.5. Correlation, PCA, and HCA Analyses

Correlation analyses were separately performed for the total fraction and absorbed fraction of elements to compare the relation between elements and quantified metabolites. Considering the total fraction of elements, only positive correlation coefficient (R) values were found, and almost all were significantly positive (Appendix A): note also that the individual metabolites were positively correlated with each other. Considering the absorbed fraction of elements, including Hg, only Hg showed significantly negative correlation with K, Ca, and AsA (Appendix A), indicating that, e.g., depletion of absorbed Hg by SNP application restores/elevates level of elements and metabolites (see respective graphs). Considering the absorbed fraction of K, Ca, and Mg, their correlation with metabolites AsA and NPT was positive, but it was negative with SP and FAA, although only some values were significant (Appendix A). It was also interesting that K and Ca were positively correlated with AsA both in total and absorbed fraction (Appendix A), indicating their potential significance under Hg excess. 

According to the PCA result, all investigated parameters (SP, FAA, NPT, and AsA) were significantly loaded into the two main components—explaining more than 97% of the variance—in which all *Evernia* samples were strongly associated within two clusters. The first principal component (PC1) explained 84.2% of the total variance, while the second (PC2) explained 13.4%. Based on the values of the component weights given in Appendix A and Figure 4, it can be seen which analytes were more dominant than the others to explain the basic components. In PC1, all investigated analytes (SP, FAA, NPT and AsA) were strongly and positively loaded into the first principal component (PC1), while FAA and, especially, NPT were dominant in PC2, leading to main separation of samples treated with or without Hg (white and black symbols, Figure 4). Furthermore, while the Hg-untreated samples (no Hg) without the presence of SNP were similar to the samples with the addition of SNP in terms of the amounts of SP, FAA, NPT, and AsA, different patterns in the FAA and NPT contents between the samples treated with different concentrations of SNP are evident for Hg-treated samples (black circles separated from squares and triangles, Figure 4), supporting the conclusion that the NO donor SNP modulates the impact of Hg toxicity. A similar structure to that obtained by PCA was found using hierarchical cluster analysis (HCA): the hierarchical cluster tree naturally divided the data into three distinct, well separated clusters (Appendix A).

## 3. Materials and Methods

### 3.1. Lichens and Experimental Design 

To detect natural Hg content, selected species of the epiphytic lichens were collected throughout the Slovak republic: *Xanthoria parietina* (at two potentially polluted sites near the metallurgic factory in the village Párnica or below the heap of the former nickel smelter in the village Dolná Streda and at one clean site in High Tatra Mountains), *Evernia prunastri* (village Staré Hory and High Tatra Mountains, both potentially clean sites), *Pseudevernia furfuracea* and *Hypogymnia physodes* (both from potentially clean sites near Ochtinská Aragonite Cave and from High Tatra Mountains), and *Usnea hirta* (from High Tatra Mountains only). Collection of lichens in the protected national park High Tatra Mountains was performed with the permission of local authorities. Lichen samples of ca. 5 g DW were collected to minimize damage of the local populations. Samples were cleaned from visible contamination (*X. parietina* and *H. physodes* carefully removed from branches using forceps to ensure samples were free of bark residues), dried at laboratory temperature, and used without rinsing for direct determination of Hg in the thalli.

For physiological experimentation with exogenous Hg (chloride) application (no Hg/−Hg or 100 µM Hg/+Hg in results), combined with a NO donor (10 or 100 µM of sodium nitroprusside/SNP, both from Sigma-Aldrich), the lichen *Evernia prunastri* (locality Staré Hory) was used. The 50 mg DW of air-dried samples (from the apical part of individual branches of thalli to ensure homogeneity) were exposed to treatments prepared in 50 mL of HEPES buffer (pH 6.5) using screw-cap inert plastic tubes (Sarstedt, Germany). Control (no Hg/no SNP) was maintained in HEPES buffer only. Samples were kept at PAR ~30 µmol m^−2^ s^−1^ with 12 h/12 h day/night regime, respectively, at 20–24 °C. Absolute dry mass of lichens was determined by weighing the sub-samples dried in an oven at 100 °C (to allow re-calculation of parameters extracted from fresh material).

After 24 h of exposure to treatments, samples were washed with deionized water and carefully dried with filter paper prior to extraction. Processing of samples for the quantification of metabolites involved manual extraction with a mortar and pestle and with the addition of inert so-called “sea sand” to obtain complete disruption [14,24]. 

### 3.2. Quantification of Hg and Minerals 

Assuming low Hg amount in natural lichen samples, direct analysis using a single purpose atomic absorption spectrometer AMA 254 (Altec Ltd., Prague, Czech Republic) based on in situ dry ashing, followed by gold amalgamation. was used. Samples were weighed in a nickel boat and analyzed under the following conditions: drying at 120 °C for 60 s; combustion in oxygen atmosphere at a temperature of approximately ~750 °C for 150 s, heating up the amalgamator to ~900 °C and waiting for 45 s, necessary for quantitative release of trapped mercury to the detection system. The absorbance of the peak area at 253.7 nm was monitored. The flow rate of the oxygen (99.5%) carrier gas was 170 mL/min. Certified reference material (BCR-679, White Cabbage, IRMM, Belgium) with 6.3 ± 1.4 µg Hg/kg was used to perform quality controls (detected value 6.25 ± 0.35 µg Hg/kg). 

To quantify Hg and macronutrients (K, Ca, and Mg) in samples from the experiment with exogenous Hg application, samples for total elements were washed with deionized water, while those for absorbed elemental content were washed with 5 mM Na_2_EDTA, to remove surface-bound ions. Thereafter, samples were mineralized in the mixture of 16% HNO_3_ and 30% H_2_O_2_ (5:2) using a microwave oven speedwave XPERT (Berghof, Eningen, Germany). Measurements were performed using an Agilent 7900 ICP-MS (Inductively Coupled Plasma—Mass Spectrometry), with technical and operating/analytical parameters as reported previously [34].

### 3.3. Microscopic Analyses

Fluorescence microscopy was carried out using Axioscop 40 microscope (Carl Zeiss, Jena, Germany) equipped with an appropriate set of excitation/emission filters. Nitric oxide (NO) was visualized using 2,3-diaminonaphthalene (DAN) and 4-amino-5-methylamino-2′,7′-difluorofluorescein diacetate (DAF-FM DA); reactive oxygen species (ROS), using CM-H_2_DCFDA and Texas Red emission filters, was used to visualize autofluorescence of chlorophyll. Samples were stained in 5–50 µM working solution prepared in 50 mM of phosphate-buffered saline (PBS, pH 6.8) in the dark, washed with respective buffer and immediately observed as reported in detail previously [14,24].

### 3.4. Assay of Metabolites

Reduced ascorbic acid and non-protein thiols were extracted in 0.1 M HCl and quantified with bathophenanthroline and Ellman’s reagent as previously stated [14]. Soluble phenols were extracted in 80% aqueous methanol and free amino acids in 60% aqueous ethanol and quantified with Folin-Ciocalteu and ninhydrin method as reported earlier [34].

### 3.5. Statistics

Statistical analyses were performed using MINITAB Release 11 (Minitab Inc., State College, PA, USA) and MATLAB^®^ R2022a (The MathWorks, Inc., Natick, MA, USA) software. Data were evaluated for normality and homogeneity using the Shapiro-Wilk and Levene tests, respectively. One-way ANOVA, followed by a Tukey’s test, was used to evaluate the significance of differences (*p* < 0.05) between –Hg and +Hg treatments, while a Student’s t-test was used to evaluate differences between −Hg and +Hg in the given treatment. The experimental data were further analyzed through principal component analysis (PCA) and hierarchical cluster analysis (HCA) using MATLAB^®^ R2022a software package (The MathWorks, Inc., USA). With respect to HCA, an agglomerative hierarchical algorithm was used, which progressively combines pairs, by measuring Euclidean distances among the clusters. The method of average linkage was selected, as it provided the highest cophenetic correlation (0.97). HCA was applied to the same input data used for PCA, namely a matrix consisting of the normalized yields obtained from each measurement.

## 4. Conclusions

The present study shows that natural Hg accumulation in the five Slovak epiphytic lichens is species-specific, but the total range did not vary extensively (~30–100 ng/g DW) and no impact of eventual pollution sources on Hg amount was observed. The physiological experiment with *Evernia prunastri* (100 µM Hg over 24 h) demonstrates the negative impact of Hg on the amount of ascorbic acid, non-protein thiols, soluble phenols, free amino acids, macronutrients (K, Ca, Mg), and chlorophyll autofluorescence, as well as a stimulatory impact of Hg on the ROS formation. Co-application of the exogenous nitric oxide donor (sodium nitroprusside) lowered Hg-induced ROS signal and absorption of Hg by thalli, as well as ameliorated Hg-induced depletion of ascorbic acid and non-protein thiols: this effect was not found for amino acids and phenols, and chlorophyll autofluorescence also remained unchanged. Data confirm that NO is a vital molecule in the lichen and provides protection against Hg-induced toxicity but its action is not universal. Due to environmental pollution with nitrogen oxides similar to NO, studying their impact on lichen diversity and physiology is a challenge for the future.

## Figures and Tables

**Figure 1 plants-12-00727-f001:**
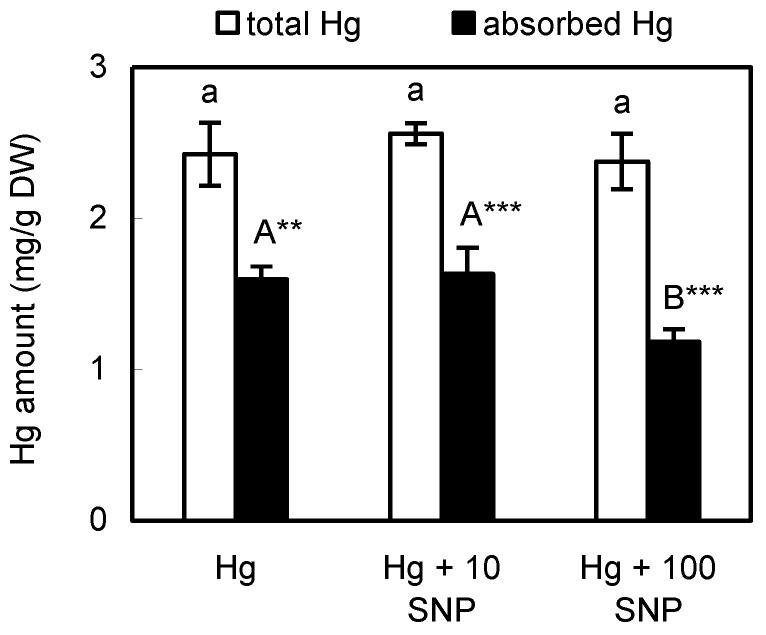
Accumulation of Hg in the lichen *Evernia prunastri* exposed for 24 h to 100 µM Hg with or without co-application of NO donor sodium nitroprusside (SNP) at a dose of 10 or 100 µM (10 SNP or 100 SNP). Total Hg content means that thalli were washed with deionized water while absorbed Hg means that thalli were washed with 5 mM Na_2_EDTA to remove surface-adsorbed Hg ions. Data are means ± SDs shown as bars (*n* = 3). Columns for a given fraction, followed by the same letter(s), are not significantly different according to Tukey’s test (*p* < 0.05). ** and *** indicate significant difference between total and absorbed fraction at 0.01 or 0.001 level of Student’s *t*-test, respectively.

**Figure 2 plants-12-00727-f002:**
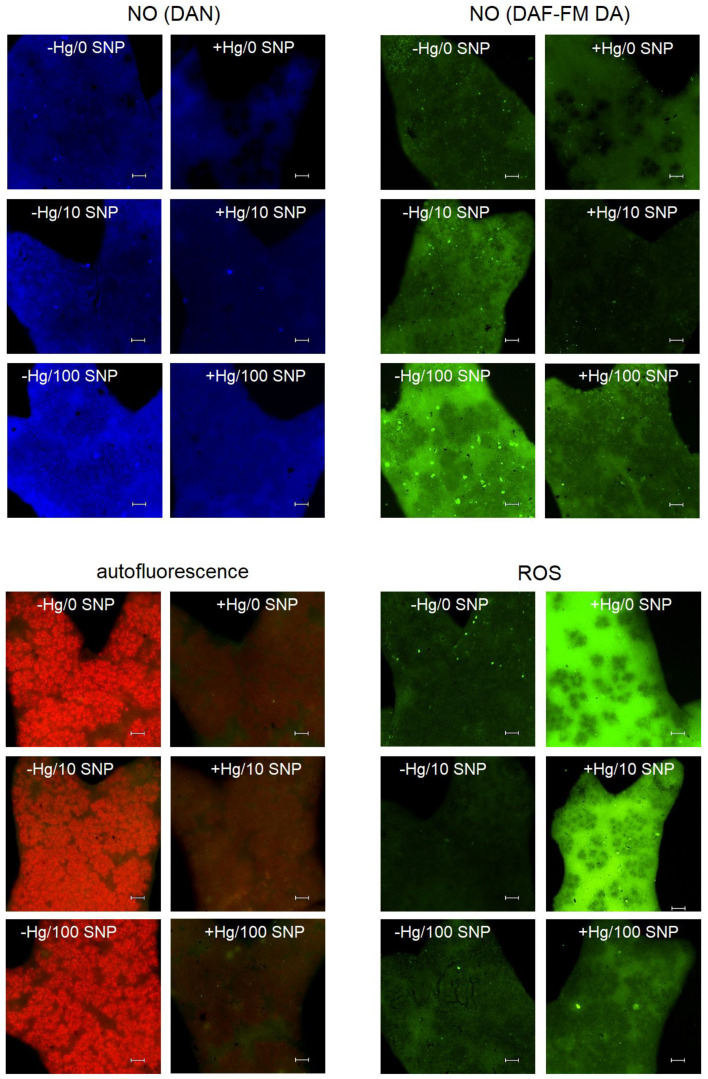
Microscopic fluorescence detection of nitric oxide (NO) using two staining reagents (DAN and DAF FM-DA), chlorophyll autofluorescence and reactive oxygen species (ROS) in the lichen *Evernia prunastri* exposed for 24 h to 100 µM Hg (+Hg on photos) or no Hg (−Hg on photos) with or without co-application of NO donor sodium nitroprusside (SNP) at a dose of 10 or 100 µM (10 SNP or 100 SNP on photos). Bar indicates 100 µm. Note that 100 µM SNP effectively increased NO formation and visibly suppressed Hg-induced ROS formation, but had no effect on Hg-induced depletion of chlorophyll autofluorescence.

**Figure 3 plants-12-00727-f003:**
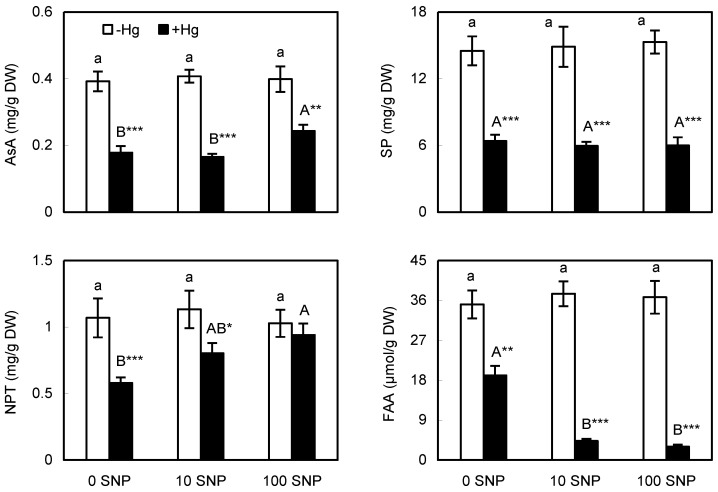
Accumulation of ascorbic acid (AsA), non-protein thiols (NPT), soluble phenols (SP), and free amino acids (FAA) in the lichen *Evernia prunastri* exposed for 24 h to 100 µM Hg (+Hg) or no Hg (−Hg) with or without co-application of NO donor sodium nitroprusside (SNP) at a dose of 10 or 100 µM (10 SNP or 100 SNP). Data are means ± SDs shown as bars (*n* = 3). Columns, followed by the same small or capital letter(s), are not significantly different according to Tukey’s test (*p* < 0.05). *, **, and *** indicate a significant difference between −Hg and +Hg in the given treatment at 0.05, 0.01, or 0.001 level of Student’s *t*-test, respectively.

**Figure 4 plants-12-00727-f004:**
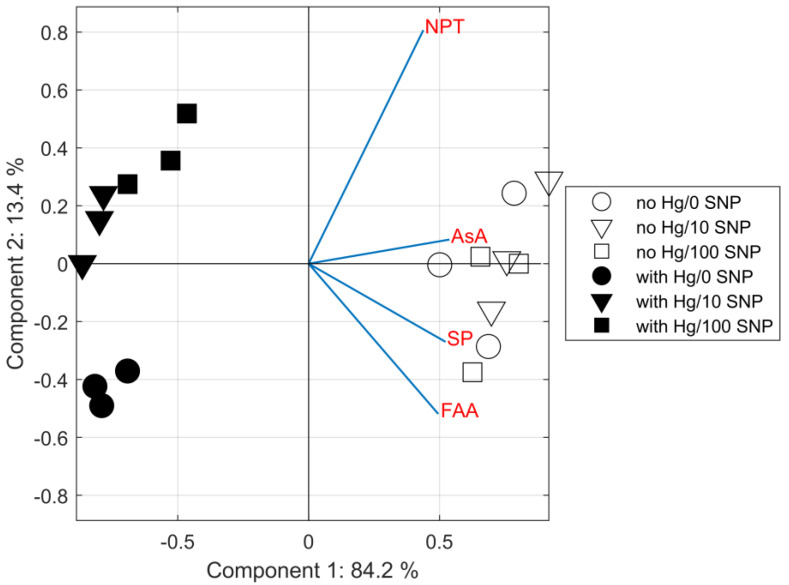
Biplot illustrating PCA analysis of metabolic parameters in the lichen *Evernia prunastri* exposed for 24 h to 100 µM Hg (with Hg) or no Hg with or without co-application of the NO donor sodium nitroprusside (SNP) at a dose of 10 or 100 µM (10 SNP or 100 SNP). NPT—non-protein thiols, AsA—ascorbic acid, SP—soluble phenols, FAA—free amino acids.

**Table 1 plants-12-00727-t001:** Content of Hg (ng/g of dry weight, DW) in selected epiphytic lichens collected in Slovakia (data are means ± SDs, *n* = 3). Values within columns, followed by the same letter(s), are not significantly different according to Tukey test (*p* < 0.05). ++ indicates polluted sites (near metallurgic factory or below heap of former nickel smelter), while + indicates presumably clean sites (forest/mountain areas). Note that all five species are available from High Tatra Mountains and differences among species are visible.

Species	Locality	Pollution Level	Hg (ng/g DW)
*Xanthoria parietina*	Párnica	++	46.7 ± 3.8 e
	Dolná Streda	++	73.6 ± 2.7 c
	High Tatra Mountains	+	55.1 ± 4.8 de
*Evernia prunastri*	Staré Hory	+	58.0 ± 2.4 d
	High Tatra Mountains	+	30.7 ± 4.1 f
*Pseudevernia furfuracea*	Ochtinská Aragonite Cave	+	108.9 ± 9.5 a
	High Tatra Mountains	+	95.3 ± 5.9 ab
*Hypogymnia physodes*	Ochtinská Aragonite Cave	+	72.1 ± 4.8 c
	High Tatra Mountains	+	106.5 ± 7.6 a
*Usnea hirta*	High Tatra Mountains	+	82.2 ± 1.9 bc

## Data Availability

The data presented in this study are available on request from the corresponding author.

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
