# Peer review of "Mercury Content and Amelioration of Its Toxicity by Nitric Oxide in Lichens"

_plants, 2023, doi:10.3390/plants12040727_

Round 1

Reviewer 1 Report

This manuscript mainly reported the physiological behavior of the epiphytic lichen, Evernia prunastri, under excess Hg absorption condition, especially amelioration of Hg toxicity by NO donor SNP.

Valuable findings were obtained and clearly supported by the experimental data.

Therefore, I recommended to accept this manuscript after minor revision in consideration of comments below.

L.34

If you mentioned “physiological studies” as a general meaning, some studies might also be referred, for example;

Nicolardi, V. et al., The adaptive response of lichens to mercury exposure involves changes in the photosynthetic machinery. Environ. Pollut. 2012, 160, 1-10.

If you means physiological studies in the presence of Hg, clear description is recommended.

L.46-47

The role of NO in lichens is less known but was reported, for example;

Weissman, L. et al. Rehydration of the Lichen Ramalina lacera Results in Production of Reactive Oxygen Species and Nitric Oxide and a Decrease in Antioxidants. Appl. Environ. Microbiol. 2005, 71, 2121–2129.

L.49-51

“Owing to absence of data related to SNP/NO effect in lichens, we tested its eventual ameliorative effect on Hg-induced toxicity in the epiphytic lichen Evernia prunastri in a laboratory experiment.”

The purpose of using NO “donor” should be clearly mentioned. That will help readers understand the explanation in L.152-156.

In case of Cd excess, as reported in [24], detectable amount of NO was generated by Ramalina so that NO scavenger is useful to check the effect of NO on ROS reduction. However, in case of Hg (in this manuscript), amount of generated NO was too small to detect significantly. In such a case, NO donor is useful, and that is the purpose of using NO donor.

L.63-64

Order of accumulation ability, Pseudevernia furfuracea > Hypogymnia physodes > Evernia prunastri, was also shown by the experimental results in the following reference.

Vannini, A. et al., Estimating Atmospheric Mercury Concentratinos with Lichens. Environ. Sci. Technol. 2014, 48, 8754-8759.

L.220-221

“It may indicate that specific interaction between Hg and SNP modulates final response”

Any references showing the interaction between Hg and SNP are recommended to be referred for this consideration.

There is a possibility of interaction with residue of SNP after donating NO. In the presence of Hg, more SNP was expected to be consumed for NO donating. That might lead to produce more residue of SNP than in the absence of Hg.

L.285-287

How much is the volume of solution? Same as volume of the plastic tube?

Author Response

Reviewer 1: This manuscript mainly reported the physiological behavior of the epiphytic lichen, Evernia prunastri, under excess Hg absorption condition, especially amelioration of Hg toxicity by NO donor SNP. Valuable findings were obtained and clearly supported by the experimental data. Therefore, I recommended to accept this manuscript after minor revision in consideration of comments below.

L.34 If you mentioned “physiological studies” as a general meaning, some studies might also be referred, for example;

Nicolardi, V. et al., The adaptive response of lichens to mercury exposure involves changes in the photosynthetic machinery. Environ. Pollut. 2012, 160, 1-10.

If you means physiological studies in the presence of Hg, clear description is recommended.

L.46-47 The role of NO in lichens is less known but was reported, for example;

Weissman, L. et al. Rehydration of the Lichen Ramalina lacera Results in Production of Reactive Oxygen Species and Nitric Oxide and a Decrease in Antioxidants. Appl. Environ. Microbiol. 2005, 71, 2121–2129.

RESPONSE: thank you for suggestion of valuable papers, they were cited and also discussed in discussion section (blue text, page 4)

 L.49-51 “Owing to absence of data related to SNP/NO effect in lichens, we tested its eventual ameliorative effect on Hg-induced toxicity in the epiphytic lichen Evernia prunastri in a laboratory experiment.” The purpose of using NO “donor” should be clearly mentioned. That will help readers understand the explanation in L.152-156.

In case of Cd excess, as reported in [24], detectable amount of NO was generated by Ramalina so that NO scavenger is useful to check the effect of NO on ROS reduction. However, in case of Hg (in this manuscript), amount of generated NO was too small to detect significantly. In such a case, NO donor is useful, and that is the purpose of using NO donor.

RESPONSE: modified as suggested (page 1-2)

L.63-64 Order of accumulation ability, Pseudevernia furfuracea > Hypogymnia physodes > Evernia prunastri, was also shown by the experimental results in the following reference.

Vannini, A. et al., Estimating Atmospheric Mercury Concentratinos with Lichens. Environ. Sci. Technol. 2014, 48, 8754-8759.

RESPONSE: thank you for suggestion of valuable paper, it is perfectly in line with our data and was discussed (blue text, page 2)

L.220-221 “It may indicate that specific interaction between Hg and SNP modulates final response”

Any references showing the interaction between Hg and SNP are recommended to be referred for this consideration. There is a possibility of interaction with residue of SNP after donating NO. In the presence of Hg, more SNP was expected to be consumed for NO donating. That might lead to produce more residue of SNP than in the absence of Hg.

RESPONSE: we found no data which we could discuss in detail but your explanation is plausible (page 7)

L.285-287 How much is the volume of solution? Same as volume of the plastic tube?

RESPONSE: yes, it is as you say (page 8)

Reviewer 2 Report

Manuscript entitled “Mercury content and amelioration of its toxicity by nitric oxide in lichens” by Jozef Kováčik et al., is of a great importance to its field. The research work focused on the determination of mercury accumulation in 5 Slovakian lichens and possibility of amelioration of Hg toxicity by NO. Toxic heavy metals, such as Hg are one of the major pollutants mainly released by human activities into the environment, which are not biologically degradable, accumulate in organisms, thereby posing a threat to human health. So, every effort that helps to explore connections, and find solutions to reduce the damages and dangers caused by heavy metals is certainly useful and necessary. The structure of the manuscript follows the requirement of the Special Issue " Role of Plants and Cyanobacteria in Environmental Resilience and Ecosystem Sustainability" of Plants.

In my opinion the ‘Introduction’ part is short. It would be important to explain and describe in more detail and deeper the importance of this study, why it is essential and necessary, issues and solutions and highlight how this study can contribute to it, etc.  The main aim of the work should be clearly determined and mentioned.

In the full text, please check:

- if the meaning of abbreviations is defined the first time they appear (some of them is highlighted in the revised text, attached);

- synchronization between the paragraphs and within paragraphs is needed (for example: Line 107-128), which makes this part readable, and will be easier to follow.

The methods were used, description of the results is correct, important, and well described.

Figures are informative. I appreciate that it was considered that the damages of the local populations should be minimized during collecting the lichen samples. Please, consider publishing 2 more figures: Fig S.1 and Fig S.2.

In the ‘Conclusion’ part I suggest explaining some future prospects, not just a summary of the research work.

Since the manuscript contains valuable, relevant, and essential information, and my suggestions do not decrease the real value of this manuscript, so I recommend for publishing it after making the required changes.

Author Response

Reviewer 2: Manuscript entitled “Mercury content and amelioration of its toxicity by nitric oxide in lichens” by Jozef Kováčik et al., is of a great importance to its field. The research work focused on the determination of mercury accumulation in 5 Slovakian lichens and possibility of amelioration of Hg toxicity by NO. Toxic heavy metals, such as Hg are one of the major pollutants mainly released by human activities into the environment, which are not biologically degradable, accumulate in organisms, thereby posing a threat to human health. So, every effort that helps to explore connections, and find solutions to reduce the damages and dangers caused by heavy metals is certainly useful and necessary. The structure of the manuscript follows the requirement of the Special Issue " Role of Plants and Cyanobacteria in Environmental Resilience and Ecosystem Sustainability" of Plants.

In my opinion the ‘Introduction’ part is short. It would be important to explain and describe in more detail and deeper the importance of this study, why it is essential and necessary, issues and solutions and highlight how this study can contribute to it, etc.  The main aim of the work should be clearly determined and mentioned.

RESPONSE: thank you, we slightly improved introduction to maintain space organization of the text (page 1-2)

In the full text, please check:

- if the meaning of abbreviations is defined the first time they appear (some of them is highlighted in the revised text, attached);

RESPONSE: corrected in the full text

- synchronization between the paragraphs and within paragraphs is needed (for example: Line 107-128), which makes this part readable, and will be easier to follow.

The methods were used, description of the results is correct, important, and well described.

Figures are informative. I appreciate that it was considered that the damages of the local populations should be minimized during collecting the lichen samples. Please, consider publishing 2 more figures: Fig S.1 and Fig S.2.

RESPONSE: these figures are presented in the supplementary material for the space reasons. We slightly modified respective paragraph so hope it is fine now (page 4)

In the ‘Conclusion’ part I suggest explaining some future prospects, not just a summary of the research work.

RESPONSE: shortly added (page 10)

Since the manuscript contains valuable, relevant, and essential information, and my suggestions do not decrease the real value of this manuscript, so I recommend for publishing it after making the required changes.

RESPONSE: thank you very much for your valuable comments.

Reviewer 3 Report

Dear authors, I found English of your paper rather spotty, sometimes it is OK, but sometimes it is hard to get through. English must be improved. Below are my suggested corrections, including problems with English:

exogenous Hg (100 μM) – what is the soluble Hg conc? Can you make us sure it is 100mkM or whatever?

an eventual side effects -- grammar

the Slovakia – I think ‘the’ is not needed.

In agreement, Evernia accumulated – it is better to say E. prunastri

is a field study – in a field study?

to 100 μM Hg accumulated 36.5 mg Hg/g DW [14], indicating much higher sorption efficiency than lichens – Do these studies use similar biomasses? Was it in both cases sorption or intracellular accumulation? Did algae grow and uptake Hg?

SNP restored 10 μM Hg-induced deletion of AsA and thiols in wild-type Arabidopsis roots – what is deletion?

I do not see the reason for Fig. 4. if you keep this figure, please write why it is needed. In it, I failed to understand the data behind individual black figures of the same shape. There are three of them of each shape. What each of them means? Why are there three of them?

Author Response

Reviewer 3: Dear authors, I found English of your paper rather spotty, sometimes it is OK, but sometimes it is hard to get through. English must be improved. Below are my suggested corrections, including problems with English.

RESPONSE: The English has been improved by translators and professional colleagues, and two other reviewers have not indicated unclear parts, so we hope it is now understandable. If not, please mark the paragraphs that are unclear and we will check it again, thanks in advance.

exogenous Hg (100 μM) – what is the soluble Hg conc? Can you make us sure it is 100mkM or whatever?

RESPONSE: exogenous means applied concentration in the form of treatment solution. Concentration was verified by ICP of course.

an eventual side effects -- grammar

the Slovakia – I think ‘the’ is not needed.

In agreement, Evernia accumulated – it is better to say E. prunastri

is a field study – in a field study?

SNP restored 10 μM Hg-induced deletion of AsA and thiols in wild-type Arabidopsis roots – what is deletion?

RESPONSE: thank you very much for careful reading, all typos were corrected.

to 100 μM Hg accumulated 36.5 mg Hg/g DW [14], indicating much higher sorption efficiency than lichens – Do these studies use similar biomasses? Was it in both cases sorption or intracellular accumulation? Did algae grow and uptake Hg?

RESPONSE: difference in biomass was only ca. 2.5-fold so lichen should contain ca. 14 mg Hg/g DW but it contained only 2 mg Hg/g DW – with respect to identical exposure conditions, we believed this comparison is at least numerically acceptable.

I do not see the reason for Fig. 4. if you keep this figure, please write why it is needed. In it, I failed to understand the data behind individual black figures of the same shape. There are three of them of each shape. What each of them means? Why are there three of them?

RESPONSE: we believe that the separation of treatments is clear (and two other reviewers had no problem with this figure) and PCA is a common method published in many other papers. Similar separation was also obtained by HCA (in supplementary materials) so the impact of Hg and subsequently of SNP is visible. 

Round 2

Reviewer 3 Report

Dear authors,

I now think your paper is mostly fine. I still think, though, that English of your abstract is not acceptable, except the last sentence of it. I dared to rewrite the first sentence, see below. Everybody has his/her style, but I believe my version is much easier to get through. However, I am not your editor.

Mercury (Hg) content measured in five epiphytic lichen species collected in Slovakia’s mountain forests ranged from 30-100ng/g DW and was species-specific, decreasing in the order Pseudevernia > Hypogymnia > Usnea > Xanthoria > Evernia prunastri.

Besides English, my questions about the abstract are:

Why some lichens have only genus names, and some fuller names? Are the measurements at polluted sites really worth mentioning in the abstract, since they are scarce and inconclusive? Does Xanthoria uptake Hg like Evernia does? I saw the difference in Fig 1.

There are some other problems I have with your abstract. For instance, “improved changes of some macronutrients” adds nothing but confusion to the reader. The other difficult places are:

NO was efficiently released from SNP as detected by two staining reagents and fluorescence microscopy and lowered Hg-induced ROS signal and absorption of Hg by thalli of Evernia prunastri — What does ‘lowered’ refer to? Did you specifically use thalli?

All these metabolites, including soluble phenols, were 19 reduced by excess Hg per se. Metabolites or their concentrations? Confusing.

Often, not much is read in a paper beyond the abstract, but your text is better than the abstract, not vice versa.

Author Response

Dear authors,

I now think your paper is mostly fine. I still think, though, that English of your abstract is not acceptable, except the last sentence of it. I dared to rewrite the first sentence, see below. Everybody has his/her style, but I believe my version is much easier to get through. However, I am not your editor.

Mercury (Hg) content measured in five epiphytic lichen species collected in Slovakia’s mountain forests ranged from 30-100ng/g DW and was species-specific, decreasing in the order Pseudevernia > Hypogymnia > Usnea > Xanthoria > Evernia prunastri.

RESPONSE: Mercury (Hg) content measured in five epiphytic lichen species collected in Slovakia’s mountain forests ranged from 30-100 ng/g DW and was species-specific, decreasing in the order Hypogymnia > Pseudevernia > Usnea > Xanthoria > Evernia prunastri (but polluted sites had no impact on Hg amount in Xanthoria). 

Besides English, my questions about the abstract are:

Why some lichens have only genus names, and some fuller names? Are the measurements at polluted sites really worth mentioning in the abstract, since they are scarce and inconclusive? Does Xanthoria uptake Hg like Evernia does? I saw the difference in Fig 1.

RESPONSE: Fig. 1 shows only data from Evernia. The full name is given for Evernia, which we used for the subsequent physiological experiment.

There are some other problems I have with your abstract. For instance, “improved changes of some macronutrients” adds nothing but confusion to the reader. The other difficult places are:

NO was efficiently released from SNP as detected by two staining reagents and fluorescence microscopy and lowered Hg-induced ROS signal and absorption of Hg by thalli of Evernia prunastri — What does ‘lowered’ refer to? Did you specifically use thalli?

RESPONSE: word “lowered” was modified to “reduced” and “improved changes of some macronutrients” was modified to “mitigated the decline of some macronutrients”

All these metabolites, including soluble phenols, were 19 reduced by excess Hg per se. Metabolites or their concentrations? Confusing.

Often, not much is read in a paper beyond the abstract, but your text is better than the abstract, not vice versa.

RESPONSE: modified as “The amount of all metabolites, including soluble phenols, was reduced by excess Hg per se.”